# Influence of Urbanization Processes on the Dynamics and Scale of Spatial Transformations in the Mazowiecki Landscape Park

**Emilia Janeczko [1], Radosław Dąbrowski [2], Joanna Budnicka-Kosior [3] and Małgorzata Woźnicka [1,***

[1] Department of Forest Utilization, Faculty of Forestry, Warsaw University of Life Sciences—SGGW, Nowoursynowska 166, 02-776 Warsaw, Poland; janeczko.emilia@gmail.com
[2] The Team of Mazowiecki Landscape Parks, 05-400 Otwock, Poland; radek@parkiotwock.pl
[3] Department of Geomatics and Land Management, Faculty of Forestry, Warsaw University of Life Sciences—SGGW, Nowoursynowska 166, 02-776 Warsaw, Poland; jbudnicka@wl.sggw.pl
[*] Correspondence: mwoznicka@wl.sggw.pl; Tel.: +48-22-5938133

**Abstract:** This paper explores how urbanization processes, since the 1950s, affected forested areas in the Mazowiecki Landscape Park and determines if these processes resulted in a significant reduction of forest. Spatial analyses, which were used to generate very detailed data on the area of forests, agricultural land, and development areas, were carried out, and the spatial directions of the changes were determined. The results indicate that, in comparison to the 1950s, in the 1990s, the forest area did not decrease, but, in fact, increased, despite a significant increase in the development area, both in the present area of the park and in its buffer zone. This was due to the fact that new buildings were constructed in mainly agricultural areas. At the same time, intensive afforestation of weak soils, mainly inland dunes, was carried out in this area. Comparing the current period to the 1990s, further dynamic growth of the development area can be observed, especially in the park's protective zone, with a simultaneous decrease in the forest area and agricultural land.

**Keywords:** urbanization; forest; spatial planning

## 1. Introduction

Urbanization is one of the most evident global changes. During the last century, rapid urban growth exerted heavy pressure on land and resources in urban areas, as well as rural areas [1,2]. Large parts of the world are highly urbanized and the majority of the world's population now lives in cities and towns [3]. Most urban residents actually live in cities that have 1–5 million people [4]. In developed nations, 80–90% of people live in cities, whereas, in the poorest nations, only 20% live in cities [3,5]. Urbanization resulted in many economic benefits for many countries, with tremendous improvements in the provision of social services to communities [6]. However, if urbanization is not adequately managed, it can lead to unfavorable environmental consequences [7]. Atmis et al. [8] highlighted the fact that urban development also leads to the exploitation of nature, resulting in an unhealthy ecology, which affects urban forest users. There are many works [9–11] devoted to the natural effects of urbanization. Suburbanization includes a decline in biodiversity, the transformation and disappearance of valuable ecosystems, and the weakening and disappearance of the natural dynamics of the environment [9,10]. Many urban areas are now confronted with overpopulation of species, for instance, deer and geese, resulting in problems such as property damage by wildlife, traffic hazards, and health concerns [10]. Urbanization, directly and indirectly, caused major shifts in species composition [11]. Coles and Millman [12] pointed out that the unprecedented urbanization,

the disproportional development between gray (settlement and built infrastructure) and green areas, and chaotic traffic systems pose great challenges to urban planners when aspiring for the provision of healthy living surroundings. Development goes hand in hand with urban deforestation, indicating increasing pressure on urban forests [8]. Suburbanization leads to an increase in investment pressure on environmentally valuable areas [13]. Strzałkowska and Hurba [14] emphasized that excessive human activity leads to the loss of value in protected areas. The scope of suburbanization influences the natural and landscape values of protected areas adjacent to the city [14]. The process of suburbanization often destroys attractive open spaces, natural ecosystems, and nonrenewable organic resources, and leads to landscape degradation [15]. In Poland, the process of urban sprawl is most visible in the largest metropolitan areas, such as Warsaw [16]. In Poland, like in other countries [8], an intensive urbanization process started in the 1950s, shortly after World War II, and was caused by internal migration from rural areas to urban areas. The next wave of intensive urbanization took place in the 1990s, and it was a consequence of the fall of communism, the implementation of free-market policy, and the reprivatization of state property. It was also a special period for the development of the real-estate market and changes in land use, as well as many social changes, which had an impact on certain spatial changes, including, above all, the processes of population movement outside the urban areas [17]. Kurek et al. [17] considered urban depopulation and settlement in suburban zones: "the most important changes in population distribution since the 1990s". Denis and Majewska [18] found that, in Poland, after the political transformation in the 1990s, the growing demand for building plots resulted in exceptionally intensive urbanization of rural areas within a radius of up to 50 km around Warsaw. Today, under the slogan "to live closer to nature", scattered single-family housing is created in the middle of fields and near forests, degrading local ecosystems [18]. For sustainable spatial management, it is important to determine the dynamics and scale of these transformations. As Samie et al. [19] noted, studying land-use change is crucial to environmental management because it influences carbon cycling, greenhouse gas emission, radiation and water budgets, and livelihoods. Land-cover and land-use change information is a very important and useful source for planners in land-use studies. Karakus et al. [20] asserted that the determination of land-use potential, by considering the capability of the land and other characteristics, provides an important data source for regional planning studies. By monitoring urban development and the conversion of fertile land and forests for different needs, city managers and planners can anticipate the potential vector(s) of city growth and, thus, counteract urban sprawl [21]. Therefore, the aim of this study was to determine the impact of urbanization on the dynamics and scale of spatial transformations in the areas adjacent to Warsaw. The Mazowiecki Landscape Park (MPK) was chosen as the study area, because it is an important part of the green belt of Warsaw.

## 2. Methods

Mazowiecki Landscape Park was established in 1986. Landscape parks established under the Nature Conservation Act [22] are one of the forms of nature protection in Poland. The idea for establishing Mazowiecki Landscape Park (MPK) was to save the existing forest complexes as an important element of the environmental and biological structure (also as the "green lungs") of the Warsaw agglomeration. The park, together with a buffer zone, covers an area of 23,710 ha in total, of which 15,710 ha is located within the park, and the remaining 8000 ha is the buffer zone. The park covers territorial units, which include districts of Warsaw, such as Wawer and Wesoła; six rural communes (in the territory of which there is no city), Celestynów, Kołbiel, Osieck, Pilawa, Sobienie-Jeziory, and Wiązowna; two municipalities (urban communes that cover only the city concerned within its administrative boundaries), Józefów and Otwock; and one urban–rural commune (including both towns and villages), Karczew (see Figure 1). Forest land covers an area of 11,858 ha within the park borders, which constitutes 75.5% of the park grounds. In the buffer zone, forests cover an area of 147,112 ha (18.4% of the buffer zone grounds). The share of private forests in the MPK area is 50%. The park is divided into two parts, the northern part, the so-called Wawerski Forest complex, and the

southern part, including the Celestynów-Otwock Forests and the vast areas of the "Bagno Całowanie" low moor. There are nine nature reserves located in the Mazowiecki Landscape Park, with three of them in the northern part.

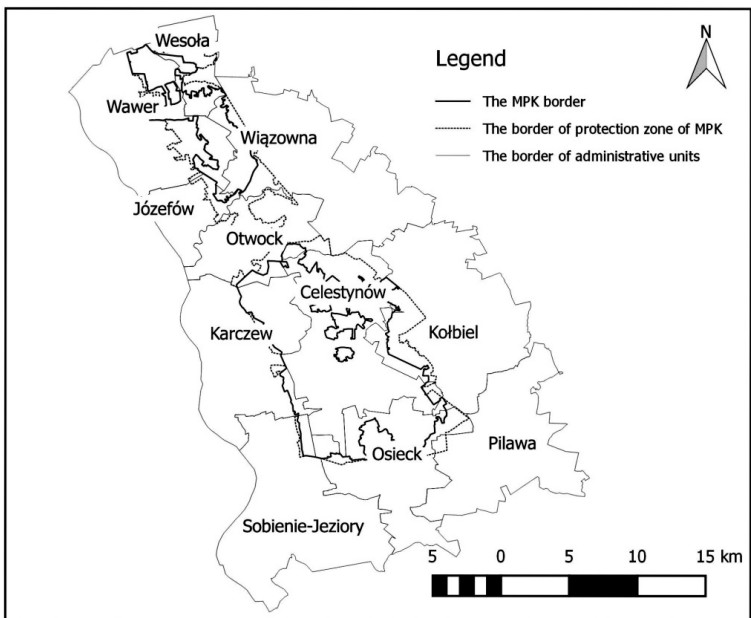

**Figure 1.** Mazowiecki Landscape Park (MPK) with the background of administration units.

This research posed the following question: "how have urbanization processes since the 1950s influenced changes in the land-use structure within the park?" In order to determine the dynamics and scale of these transformations, spatial analyses based on the following materials were performed:

- Analog aerial photographs from 1955–1960, black and white, made vertically (approximately 250 pieces). Photos were taken from the park's archives.
- Diapositive aerial photos from 1992 in the form of plates—red color, made vertically (approximately 100 pieces). Photos were taken from the park's archives.
- Ortophotomap from 2010 with real geographical representation, multi-colored, WMS overlays (Web Map System) with a current land and building register (as of December 2017).

The starting point for spatial analyses was the transformation of analog materials (photos and diapositives) into an electronic version and giving them a real geographic mapping. Quantum geographic information system (GIS) software (QGIS 2.14 Essen) and a scanner with an additional light source were used for this purpose. In QGIS, the spatial coverage of polygons—forests, agricultural land, and building objects—was determined on consecutive vector layers. The layers of polygons, created as a result of preparatory works, enabled the generation of figures concerning forest areas, agricultural land, and development areas in the three analyzed time periods, 1955, 1992, and 2017. The comparison of these data was the key to assess the trends in land transformation in the park and its protection zone and also within the borders of individual administrative units (Warsaw, urban communes, rural communes, and urban–rural communes). The results are presented in tabular form and on maps.

## 3. Results

In 1955, the development area within the park covered 63.89 ha, which represented 0.4% of the area. Specifically, 43.75 ha (68.5%) of the development area was in the rural communes, 9.89 ha (15.5%) within the borders of Warsaw, 8.21 ha (12.9%) in the municipalities, and 2.04 ha (3.2%) in the urban–rural area. The development area in the buffer zone covered 367.92 ha (4.6% of the buffer zone), of which 179.41 ha (48.8% of the development area in the buffer zone) was the built-up area

in the rural communes, 114.56 ha (31.1%) in the urban communes, and 73.95 ha (20.1%) in Warsaw. There were no development areas within the urban–rural commune (see Table 1). At the same time, the nemorous region of the forest area within the current park grounds was 65.5% and 19.8% in the buffer zone. The forest area within the park grounds was 10,291.72 ha and 1587.31 ha in the buffer zone. The majority of the forest area was in the current park borders; 6343.99 ha (61.6%) was located in the rural communes, 1809.69 ha (17.6%) within the city limits of Warsaw, 1737.43 ha (16.9%) in the urban–rural communes, and 400.61 ha (3.9%) in the municipalities. In the park's buffer zone, 765.17 ha (48.2%) of the forest area was located in the municipalities, 576.81 ha (36.3%) within the borders of the rural communes, and 245.33 ha (15.5%) in Warsaw. In the urban–rural commune, there were no forests in the park's buffer zone (see Table 2). The agricultural land within the park covered 4993.89 ha, which represented 31.8% of the area. Of the agricultural land, 3819.2 ha (76.4%) was in the rural communes, 722.01 ha (14.5%) within the borders of Warsaw, 89.11 ha (1.8%) in the municipalities, and 363.57 ha (7.3%) in the urban–rural area. The agricultural land in the buffer zone covered 5612.06 ha (70.1% of the buffer zone), of which 3665.92 ha (65.3% of the development area in the buffer zone) was in the rural commune, 628.03 ha (11.2%) in the urban commune, 1167.06 ha (20.8%) in Warsaw, and 151.05 ha (2.7%) within the urban–rural commune (see Table 3).

**Table 1.** A comparative analysis of building area in 1955, 1992, and 2017.

| Administration Area | Building Area (ha) | | | | | | Changes in Area (ha) | | | |
| | | | | | | | Changes in Area (%) | | | |
| | 1955 | | 1992 | | 2017 | | 1955–1992 | | 1922–2017 | |
| | P * | PZ * | P | PZ | P | PZ | P | PZ | P | PZ |
|---|---|---|---|---|---|---|---|---|---|---|
| Warsaw | 9.89 | 73.95 | 21.02 | 219.73 | 42.43 | 349.46 | 11.13 | 145.78 | 21.41 | 129.73 |
| | | | | | | | +112.5 | 197 | +101.9 | +59.0 |
| Urban communes | 8.21 | 114.56 | 9.28 | 216.35 | 10.3 | 277.62 | 1.07 | 101.79 | 1.02 | 61.27 |
| | | | | | | | 13 | +88.8 | +11.0 | +28.3 |
| Rural communes | 43.75 | 179.41 | 62.53 | 357.4 | 85.35 | 525.52 | 18.78 | 177.99 | 22.82 | 168.12 |
| | | | | | | | +42.9 | +99.2 | +36.5 | +47.0 |
| Urban–rural communes | 2.04 | 0.0 | 2.55 | 1.45 | 3.42 | 21.33 | 0.51 | 1.45 | 0.87 | 19.88 |
| | | | | | | | 25 | - | +34.1 | over 13× |
| Altogether | 63.89 | 367.92 | 95.38 | 794.93 | 141.5 | 1173.93 | 31.49 | 427.01 | 46.12 | 379 |
| | | | | | | | +49.3 | +116.1 | +48.4 | +47.7 |

\* P—the park; \* PZ—the protection zone.

**Table 2.** A comparative analysis of forest area in the analyzed periods of time.

| Administration Area | Forest Area (ha) | | | | | | Changes in Area (ha) | | | |
| | | | | | | | Changes in Area (%) | | | |
| | 1955 | | 1992 | | 2017 | | 1955–1992 | | 1922–2017 | |
| | P * | PZ * | P | PZ | P | PZ | P | PZ | P | PZ |
|---|---|---|---|---|---|---|---|---|---|---|
| Warsaw | 1809.69 | 245.33 | 2362.5 | 830.15 | 2362.41 | 752.07 | 552.81 | 584.82 | 0.09 | 78.08 |
| | | | | | | | +30.5 | +238.3 | −1 | −9.5 |
| Urban communes | 400.61 | 765.17 | 489.28 | 1020.37 | 489.1 | 971.69 | 88.67 | 255.2 | 0.18 | 48.68 |
| | | | | | | | +22.1 | +33.3 | −1 | −4.8 |
| Rural communes | 6343.99 | 576.81 | 7371.65 | 682.19 | 7470.46 | 691.44 | 1027.66 | 105.38 | 98.81 | 9.25 |
| | | | | | | | +16.2 | +18.3 | +1.3 | +1.3 |
| Urban–rural communes | 1737.43 | 0.0 | 1751.85 | 47.57 | 1745.28 | 48.89 | 14.42 | 47.57 | 6.57 | 1.32 |
| | | | | | | | +0.8 | - | −0.4 | +2.7 |
| Altogether | 10,291.72 | 1587.31 | 11,975.28 | 2580.28 | 12,067.25 | 2464.09 | 1683.56 | 992.97 | 91.97 | 116.19 |
| | | | | | | | +16.4 | +62.6 | +0.7 | −4.5 |

\* P—the park; \* PZ—the protection zone.

**Table 3.** A comparative analysis of agricultural land in the analyzed periods of time.

| Administration Area | Agricultural Land (ha) | | | | | | Changes in Area (ha) | | | |
|---|---|---|---|---|---|---|---|---|---|---|
| | | | | | | | Changes in Area (%) | | | |
| | 1955 | | 1992 | | 2017 | | 1955–1992 | | 1922–2017 | |
| | P* | PZ* | P | PZ | P | PZ | P | PZ | P | PZ |
| Warsaw | 722.01 | 1167.06 | 164.14 | 364.88 | 131.93 | 222.27 | 557.87 | 802.18 | 32.21 | 142.61 |
| | | | | | | | −77.3 | −68.7 | −19.6 | −39.1 |
| Urban communes | 89.11 | 628.03 | 1.93 | 260.76 | 0.49 | 224.42 | 87.18 | 367.27 | 1.44 | 36.34 |
| | | | | | | | −97.8 | −58.5 | −74.6 | −13.9 |
| Rural communes | 3819.2 | 3665.92 | 2807.82 | 3376.98 | 2645.75 | 3190.25 | 1011.38 | 288.94 | 162.07 | 186.73 |
| | | | | | | | −26.5 | −7.9 | −5.8 | −5.5 |
| Urban–rural communes | 363.57 | 151.05 | 346.33 | 101.73 | 342.77 | 80.6 | 17.24 | 49.32 | 3.56 | 21.13 |
| | | | | | | | −4.7 | −32.7 | −1 | −20.8 |
| Altogether | 4993.89 | 5612.06 | 3320.22 | 4104.35 | 3120.94 | 3717.54 | 1673.67 | 1507.71 | 199.28 | 386.81 |
| | | | | | | | −33.5 | −26.9 | −6 | −9.4 |

* P—the park; * PZ—the protection zone.

In 1992, development in the park area proceeded mainly in non-forest areas, used for agriculture. In comparison to the previous period, there was an increase of 42.9% in the development area in the rural commune areas within the park grounds (see Table 1). In the buffer zone of the park, it is possible to observe a dynamic increase in development area primarily in the Warsaw area, almost double compared to the previous period. In the urban communes, the development area increased by 88.8%, while, in rural areas, it increased by 99.2%. Development appeared in the urban–rural commune area (+1.45 ha) also. The area of forest in this period was 11,975.28 ha, which means that there was an additional 1683.56 ha of forest in the park. Similar to the park, the forest cover in the buffer zone also increased by 32.3%, which means that, up to 1955, forest expanded by 992.97 ha. In the buffer zone, the forest area totaled 2580.28 ha. The increase in forest area was the result of an area of intensive afforestation, mainly dunes, carried out in the 1960s, primarily in the northern part of the park in the districts of Warsaw, in the urban community of Otwock, and to a lesser degree in the southern part, in the Celestynów rural commune. The situation was similar in the park's buffer zone (see Figure 2). As can be seen in Table 2, in the part of the park which is in Warsaw, forest area increased by 552.81 ha, which means a 30.5% growth compared to the previous period. In the Warsaw part of the buffer zone, 584.82 ha of new forest also appeared, which indicates an increase of double in the forest area. A significant increase in the forest area, both in the park and its buffer zone, in comparison to the previous period, also took place in municipalities. In the municipalities in the park, the area increased by 88.67 ha (an increase of 22.1% compared to 1955) and by 255.2 ha in the buffer zone (an increase of 33.3%). Simultaneously, along with the increase of the forest area, the process of paving over this area took place. A positive change balance in the forest area indicates that urbanization processes occurred in a rational manner, without depletion of forest resources. However, with the increase in forest area, there was a significant decrease in the area of agricultural land. In comparison to 1955, 1673.67 ha of agricultural land disappeared in the park and 1507.71 ha in its buffer zone. In the park, the agricultural land in the Warsaw districts decreased by 557.87 ha (77.3%), by 87.18 ha (97.8%) in the urban communes, by 17.24 ha (4.7%) in the urban-rural communes, and by 1011.38 ha (26.5%) in the rural communes. In the buffer zone, the agricultural land belonging to Warsaw decreased by 802.18 ha (68.7%). There was also a decrease in agricultural land by 367.27 ha (58.5%) in the urban communes, by 288.94 ha (7.9%) in the rural communes, and by 49.32 ha (32.7%) in the urban–rural communes (see Table 3).

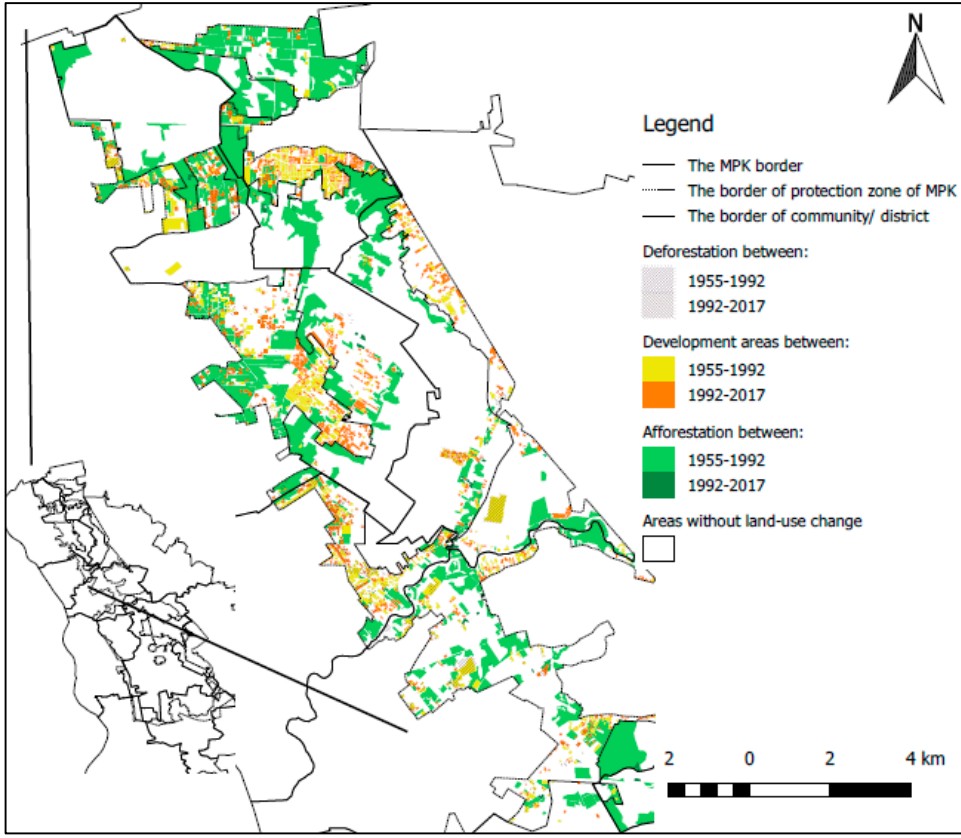

**Figure 2.** Buildings, forest area, and agricultural land in the north part of MPK.

Currently, the area of development in the park increased by 46.12 ha (48.4%) in comparison to 1992. As can be seen in Table 1, the greatest growth was in Warsaw, by 101.9% (21.41 ha). In the rural communes, there was a growth of development of 22.82 ha (36.5%), and in the urban–rural communes of 0.87 ha (34.1%). On the other hand, the development in the park's buffer zone increased by 379 ha (47.7%), including the area of Warsaw where it increased by 129.73 ha (59%), in the urban communes by 61.27 ha (28.3%), in the rural communes by 168.12 ha (47%), and in the urban–rural communes by 19.88 ha (over 13 times more). The forest area of the park today is 76.8% and that of the buffer zone is 30.8%. In comparison to 1992, a total of 91.97 ha of forest appeared in the park, while 116.19 ha of forest disappeared in its buffer zone. In the park, in comparison to the previous period, the forest area in the part consisting of the Warsaw districts decreased by 0.09 ha (1%), by 0.18 ha (1%) in the urban communes, and by 6.57 ha (0.4%) in the urban–rural communes. In the buffer zone, the forest area belonging to Warsaw decreased by 78.08 ha (9.5%). There was also a decrease in forest area by 48.68 ha (4.8%) in the urban communes. Reduction in forest area, especially in the "urban" part of the forest (Warsaw, urban communes), with a simultaneous increase in the area of development within the borders and in the buffer zone of the park, indicates that, between 1992 and 2017, some forest areas were transformed into buildings. The forest area was reduced for the benefit of settlement and industry. On the other hand, the forest area in the borders of MPK in the rural communes increased by 98.81 ha (1.3%). In the buffer zone, the area of forest in the rural communes also increased by 9.25 ha (1.3%), and in urban–rural communes increased by 1.32 ha (2.7%). As shown in Figure 3., a significant part of the agricultural land in the southern part of the park was afforested. At the same time, in comparison to 1992, a total of 199.28 ha of agricultural land disappeared in the park and 386.81 ha in its buffer zone (see Table 3). In the park, the agricultural land in the Warsaw districts decreased by 32.21 ha (19.6%), by 1.44 ha (74.6%) in the urban communes, by 162.07 ha (5.8%) in the rural communes, and by 3.56 ha (1%) in urban–rural communes. In the buffer zone, the agricultural land belonging to Warsaw decreased by 142.61 ha (39.1%). There was also a decrease in agricultural land by 36.34 ha (13.9%)

in the urban communes, by 186.73 ha (5.5%) in the rural communes, and by 21.13 ha (20.8%) in the urban–rural communes.

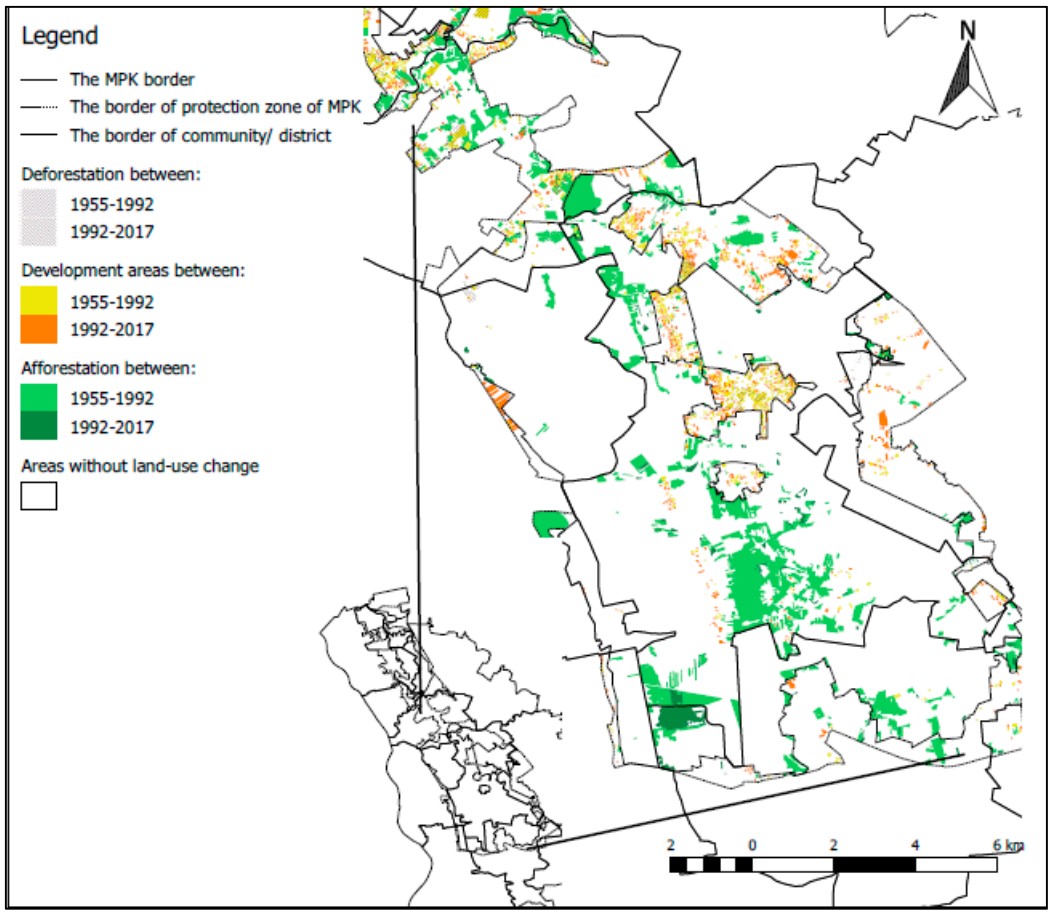

**Figure 3.** Buildings, forest area, and agricultural land in the south part of MPK.

## 4. Discussion

The results obtained indicate that, up to 1992, the development in the park and its buffer zone did not cause any significant reduction of the area occupied by forests. Along with the development process, there was an increase in the forest area. The development progressed mainly in the areas used for agricultural purposes, within the reach of rural communes, i.e., in the southern part of the park and in the area of Warsaw. Both the buildings and afforestation, as shown by the conducted research, were the result of the transformation of agricultural land. Within the boundaries of the park, the vast majority of agricultural land is designated for forestry purposes. The loss of agricultural land in the buffer zone was, to a greater extent, connected with the development of the area and related infrastructure. Połаwski [23] indicated that, in the 1950s and 1960s, the most intensive increase in forest area throughout Poland occurred at the cost of arable land, meadows, pastures, and wastelands. This is confirmed by Bański [24], who showed that the increase in forest area was related to the afforestation of soils with low agricultural suitability. Such soils dominate around Warsaw. Since 1992, in the MPK area and its buffer zone, it is possible to notice the phenomenon of the progressive expansion of development (settlement and industrial), which enters the forest areas, as well as those used for agricultural purposes. The greatest reduction in agricultural land area, and at the same time a decrease in forest area, was observed in the urban communes and in the Warsaw. Studies of Śleszyński et al. [25] showed that, in 2012–2014, there was an 8% increase in forest areas in Poland (5000 ha), which were designed for non-forest purposes. In 2012, the change in the purpose of these lands concerned 64,900 ha,

while, in 2014, it concerned 70,000 ha. The pressure to meet investment needs is constantly increasing, which requires new areas to be designated for housing estates [26].

Many studies [27–30] showed that deforestation is reduced in protected areas. In principle, protected areas stem forest clearing and degradation within their borders by restricting land-use change and extractive activities [31]. This is also the case of Mazowiecki Landscape Park. Within its boundaries, an increase in forest area was observed. On the other hand, in the buffer zone, especially in the northern part, within the range of Warsaw and urban communes such as Józefów and Otwock, the process of deforestation is progressing.

The research carried out showed that, in comparison to the park area, the development in the buffer zone is progressing more dynamically, also at the cost of forest land. The buffer zone is a protection zone in which the allocation of forest land for non-forest purposes (also intended for construction purposes) is much easier than in the park due to the lack of restrictive prohibitions, such as those applied in the park. Forest area transformation for purposes related to development in the park is less clear because of stricter legal regulations. Article 16 paragraph 6 of the Act of 16 April 2004 on Nature Conservation [22] leaves forests, agricultural land, and other properties in the landscape park area for economic use. The Protection Plan, established in 2004 by the Voivode of Mazovian Voivodeship Regulation [32], has a significant impact on the restrictions of the development in the park. The studies carried out showed that development pressure is not limited to forests in urban areas. The example of MPK shows that development is as intensive in the suburban area as in the rural areas, located mainly in the southern part of the park, within 30 km of Warsaw.

Progressive development in the suburban zone and the accompanying decrease in the forest area results from the fact that the park is located within the Warsaw agglomeration. The proximity of the capital city is the reason why the towns located in its area are commonly referred to as the "bedroom of Warsaw". Areas around cities, including Warsaw, are attractive for residents of large urban centers, as they are often characterized by a higher quality of the natural environment and lower real-estate prices [33]. They are also more attractive in terms of investment, and they have a wider range of uses for various functions preferred by investors [34]. It should be acknowledged that, along with Kacprzak and Staszewska [35], the most attractive housing estates were built in the vicinity of areas of high natural value, which may cause their degradation. The development within the park and its buffer zone, as shown by the research carried out, is also taking place in rural communes.

Patterson et al. [10] pointed out that increased urbanization pressure on rural areas poses new challenges for wildlife management. Preventing free migration of wild animals, disappearance of habitats of rare and protected plant and animal species, and reduction and fragmentation of the area of valuable plant communities in open areas (meadows, peat bogs, grasslands, fields) are listed as the main problems related to the progressing anthropopressure in the Mazovian Landscape Park [32]. The method of elimination or reduction of these threats is, inter alia, promotion of extensive agriculture and implementation of agri-environmental programs and conduction of a rational afforestation program, taking into account preservation of valuable open areas.

Urbanization leads to many conflicts against the human–environment infrastructure background [36]. Urban sprawl into rural areas causes many environmental and spatial problems, e.g., destruction of forest areas, impoverishment of soil [37], the emergence of new forms of land use [38], and excessive tourist and recreational pressure [39]. Intensive development of recreational infrastructure was observed in the Masovian Landscape Park and its buffer zone since the beginning of this century [40]. In the years 2007–2013, 77 km of horse-riding trails with resting places were created here, and nine educational trails were established. In 2009, the Forest Education Center was opened in the rural commune of Celestynów (southern part of the park). The development of recreational infrastructure is aimed at relieving the northern part of the park, where investment processes are particularly visible.

Various groups of stakeholders, such as rural people and urbanites, expect different things from the forests. When such differences are not balanced, forest resources can be damaged [8]. Sutherland

and Nash [41] drew attention to another problem concerning urbanization in rural areas. They think that, compared to rural agrarian societies, urbanized societies provide less stable ways to understand the relationship between humans and wildlife. The rational spatial policy of the commune (city) authorities should be conducive to limiting conflicts in space connected with urban sprawl and to reduce the area of forest land. Mutuga [42] considers that protected areas are under threat and subject to pressure due to urbanization. The underlying problem is a lack of integrated planning, because urban planners and park managers are divorced from each other's work. As Hersperger et al. [43] pointed out, the science of land-use change so far pays little attention to spatial policy and planning in the urban landscape, despite the widely accepted assumption that planning influences urban change. As postulated by Gawroński and Popławski [44], spatial management in ecologically protected areas should limit the possibility of local governments implementing their own spatial concepts. The planning and management of land use should be an attempt to achieve a land-use configuration that will balance stakeholder and environmental needs [45]. At the moment, the challenge for urban planners, as Atmis et al. [8] pointed out, is a spatial development process that does not cause deforestation. However, according to Feltynowski [46] and Szczepańska and Szczepański [39], even appropriate spatial development planning and prepared environmental documentation are often insufficient to respond properly to the effects of changing the forest land use for other purposes. The report of the Supreme Chamber of Control [47] confirms the opinion on weak tools of space planning and control in Poland mentioned above.

## 5. Summary

Forests play a crucial role in sustaining life on earth. Despite the increasing awareness of the importance of these ecosystems, global deforestation rates remained alarmingly high over the past decades (Food and Agriculture Organization of the United Nations, FAO) [48]. The main drivers of global deforestation are linked to wood extraction, infrastructure extension, population growth, and the expansion of agriculture [49]. Ferretti-Gallon and Busch [50] suggested that deforestation is generally lower in high, steep, and wet areas, while it is higher in areas where forests are closer to roads and urban areas. The development of buildings, especially in urban agglomerations, as shown in this study, leads not only to deforestation of land, but also to the disappearance of agricultural land. The conducted research shows that the urbanization process which lasted from the 1950s almost until the 1990s did not result in any shrinkage of forest resources in the MPK and its protection zone. In this period, the loss of agricultural land within the MPK boundaries was connected with afforestation. In the protection zone, 70% of the area previously used for agricultural purposes was afforested, while the remaining part was occupied by buildings and accompanying infrastructure. The process of transforming forests into development areas was evident since the 1990s, especially in the area of Warsaw and urban communes located in the northern part of the park. This applies to forests both in the park and in its buffer zone. At the same time, we are observing a further reduction in the area used for agricultural purposes. This situation leads to the separation of forests in the northern part of the MPK and the disappearance of ecological corridors connecting the northern and southern parts of the park. This situation suggests that the creation of protected areas in the vicinity of cities is not a sufficient way to counteract urbanization in areas of high natural values. Changes in land use caused by urbanization will pose significant challenges for wildlife agencies and policies in the future, especially with respect to the nature of decision-making processes. Urban sprawl, converting valuable agricultural areas and greenfield sites into low-density housing or for commercial building areas, is the most urgent problem of urban planning [51]. Studies on land change in urban regions, and the dynamics and scale of these transformations provide planners with important information to sustainably manage these areas. This is also important because urban land change includes many new urban–rural spaces functionally tied to the city and has many impacts on rural areas and, hence, deserves more attention in land-change science [43].

While the transformation of agricultural and forest land into residential areas is an urgent problem for urban planning, it should be noted that Polish law in this area—the Act of 3 February 1995 on the protection of agricultural and forestry lands [52]—limits the transformation of such land for non-forest purposes. The change of forest land use requires the permission of specific authorities (environmental minister or marshal of the voivodeship) during the preparation of the local spatial development plan. Specific legal provisions, as well as policies favoring afforestation of land, reduce the degradation of forest land that is particularly protected.

**Author Contributions:** Conceptualization, E.J., R.D. and M.W.; formal analysis, R.D. and J.B.-K.; methodology, E.J., J.B.-K. and M.W.; resources, E.J. and J.B.-K.; visualization, R.D.; writing—original draft, E.J.; writing—review & editing, M.W.

**Funding:** This research received no external funding.

**Conflicts of Interest:** The authors declare no conflict of interest.

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
