# Peer review of "Influence of Urbanization Processes on the Dynamics and Scale of Spatial Transformations in the Mazowiecki Landscape Park"

_sustainability, doi:10.3390/su11113007_

Round 1
Reviewer 1 Report
In the current version, the article is more transparent and logical.
I don't feel qualified to judge about the English language and style, however I found some dubious entries (e.g. lines 164-165, 230-231)
The paper does not refer to the Act of February 3, 1995 on the protection of agricultural and forest land (ie, Act of Laws of 2017, item 1161). A smaller scale of exclusions of forest land from production and their purpose for other purposes also results from a more restions approach to the protection of forest land. It is worth discussing this in the discussion.
Author Response
Dear Sir/Madam
Thank you very much for drawing our attention to an important act of legislation concerning the protection of agricultural and forestry land in Poland (the Act of February 3, 1995 on the protection of agricultural and forest land (ie, Act of Laws of 2017, item 1161). We referred to the importance of this document in the Conclusion.
Reviewer 2 Report
Some wording and punctuation mistakes should be removed:
- The tables 1. and 2. have been incorrectly signed. The headings of tables 1., 2., 3. ought to be unified.
- The spelling of the tables’ names in text should be fixed: small or capital letter (in 160 line and 180).
- There are some mistakes with use or non-use of dashes (in lines: 39,52, 53,227), the slash (in 71 line), the brackets (in 89 line) and the comma (in 183 line).
- There is not measure (ha) in brackets in 166 line.
- There are some incorrect or incomprehensible phrases or sentences (in lines: 229, 275-277, 311).
- The sentence in 256-257 is very near to the next one in 258-259. It can be deleted.

Author Response
Dear Sir /Madam
Our oversight was to change the names of the table. Thank you for drawing attention to this fact.
In the improved version we fixed this mistake. We also fixed some mistakes with „use or non-use of dashes, the slash, the brackets, and the commas”. In the text, as suggested by the Reviewer, we have crossed out a sentence that was meaningfully duplicated in the next part of the text. When it comes to the issue of incorrect or incomprehensible phrases or sentence we've decided to direct our text to professional English editing service (MDPI) which provides an English editing service checking grammar, spelling, punctuation and some improvement of style.
Reviewer 3 Report
Please see the attached.

Author Response
Dear Sir/ Madam
We've studied the impact of urbanization processes on the dynamics and scale of spatial transformations in the Mazowiecki Landscape Park on a long-term scale. We wanted to know the scale of this phenomenon and the spatial direction of changes resulting from the development of buildings. For us foresters it is important to recognize this phenomenon and determine to what extent the forests in the surroundings of Warsaw, the capital of Poland, are protected against urbanization pressure. We believe that with our findings planners and architects will be able to suggest useful strategies or treatments for each target area including forested, rural or urban areas dealt with in this study.
As for the comments relating to grammar, of course we will use the reviewer's suggestion and direct our text to professional English editing service
We would like to thank you very much for the relevant, constructive comments included in the reviews of our text.
Round 2
Reviewer 3 Report
Please see the attached.

Author Response
Dear Reviewer,
thank you very much for the relevant, constructive comments contained in the reviews of our text.
Referring to the first comment;
We would like to thank you very much for drawing our attention to the values - digits in table 3 that summarise our reflections on the changes in the agricultural land. In the table 3 the total area of land that changed its character from agricultural to other in the protection zone was wrongly given. That error should not be present, especially at this stage of the procedure. The more we're grateful for finding this mistake. Finally, in the protection zone, the area of agricultural land decreased by 386.81 ha. The correct value is placed in the table and in the text.
Referring to the second comment;
We fully agree with the reviewer's remark that documenting the changes referring to ecological issues requires more attention. We are also convinced that changes in land use affect ecological structures and functions. But on the basis of the collective materials we are not able to determine the to fully quantify this impact. Our research is only one step in this direction.
Referring to next comment:
We are aware that the tables are extensive. Adding more columns may further reduce their legibility. In one of the first versions of the manuscript, we adopted a different layout - three tables, each concerning a different research period, containing information on the total area of individual administration units in the view of a park / protective zone. But this arrangement did not satisfy us because it did not give us a chance to compare particular issues (buildings, forests and agriculture changes). Therefore, we would like to maintain the current version of the tables.
We improved all decimal point and comma.
In terms of comments on the English language, we decided to use the professional English language editing service offered by Sustainability and we have nevertheless consulted the text also with a native speaker.
Round 3
Reviewer 3 Report
Please see the attached.

Author Response
Dear Reviewer,
thank you very much for the relevant, constructive comments contained in the reviews of our text.
We have taken into account all of the reviewer's comments. We have improved both the tables and the maps and we hope they are now more readable.
This manuscript is a resubmission of an earlier submission. The following is a list of the peer review reports and author responses from that submission.
Round 1
Reviewer 1 Report
The authors undertook a very important issue in the research. Warsaw is the capital of Poland and the most dynamically developing city. It has a very strong impact on the urbanization of neighboring rural areas (used in agriculture and forests). Addtionally, in this case the research concerned of the area covered by the legal form of nature protection.
Detailed comments:
- the abstract is too long,
- the article will be more readable, if in table 1, the data of forest area will also be provided for 2017 (if it possible). This data we can find in figure 7 - it is OK (the structure of the article is correct and logical). If autors have another opinion (in case of table 1) I will accept,
- I suggest increasing the literature review, especially discussion. It may be worth looking what are the trends in other countries. What is the pressure to develop of areas covered by the legal form of nature procetion? What are the restrictions and procedures?
- references require formatting in accordance with the publisher's guidelines.
Final conclusion: The article has potential
Reviewer 2 Report
General comments
Title of the article is broader than its content. Research hypotheses are wrongly formulated and they have not been proven. Authors have established only that the forests area of MPK increased in comparison to 1992. The characteristics, reasons and consequences of the phenomenon were described (not proven) very superficially.
The results are only partially well described. The spatial distribution of forests changes is omitted. The discussion is too general and limited without any quotations of other studies on forests of landscape parks in Poland regarding different aspects of the urbanization. The conclusions are partly unfounded.
Specific comments
Abstract
- There is too detailed information (about the MPK, the characteristics of urbanization in Poland and software) which should be moved to Introduction.
- There is not information about applied methods. The QGIS 2.14 is the tool, not the method.
- The article is not answer to the questions in 21-22 lines. It describes and confronts the changes of development and forest areas in specified periods (“the dynamics and scale of land transformation” in line110-111).
- The last sentence ( 31-33 lines) is not related to the article content.
- It would be better to add the word Poland after Warsaw in line 15.
Introduction
General description of the urbanization process is not enough for the introduction. It would be appropriate to add more detailed information about the national and local context (the characteristics of the urbanization process in landscape parks in Poland; features of the MPK).
Methods
- There is no an explanation why 1955 and 1992 were adopted for comparisons.
- It is necessary to provide sources for aerial maps from 1995 and 1992.
- Is the described research part of a larger study? It must be explained.
- The subtitle Study Design is unnecessary.
- Figures 2-4 do not provide any information, they could be removed.
Results
- The Authors have to define the administration areas from Table 1. and Table 2. and assign them to the names of the counties.
- Why is the Table 1. missing data for 2017?
- Why didn’t the study compare changes in the area of arable land too?
- It will be interesting to know what part of the farmland area was transformed into forests or what part of the forest was grubbed up for any functions.
- The Figures 5 -6 are illegible.
- The changes in spatial distributions of the MPK forests have to be described and assessed.
- There is no measure and year in the signature under the Figures 7.
Discussion
The statements in the Discussion part are unfounded. We must believe the Authors in the case of (for example) the cost of forest land (221 line) or factors of investment attractiveness of the areas around Warsaw (244-249 line).
- The sentence in 224-227 lines is unclear.
- There is repeated content in 234-237 lines.
Conclusions
The conclusions are too short and too little related to the results and their discussion (only two sentences in 275-278 lines relate to them).

Reviewer 3 Report
The introduction section (literature review) has very limited information. There is not much information on similar studies and the methods they used in other published articles. The gap in what has already been researched and what is trying to be accomplished in this study is greatly missing. Even though the question you are trying to study regarding the impact of urbanization on the area covered by forests and the discussions are significant, there are no innovative or novel tools or methods to fulfill the objective of your study. The study design can be improved and elaborated too.